# Latent representations in hippocampal network model co-evolve with behavioral exploration of task structure

Ian Cone [1] ✉ & Claudia Clopath [1]

To successfully learn real-life behavioral tasks, animals must pair actions or decisions to the task's complex structure, which can depend on abstract combinations of sensory stimuli and internal logic. The hippocampus is known to develop representations of this complex structure, forming a so-called "cognitive map". However, the precise biophysical mechanisms driving the emergence of task-relevant maps at the population level remain unclear. We propose a model in which plateau-based learning at the single cell level, combined with reinforcement learning in an agent, leads to latent representational structures codependently evolving with behavior in a task-specific manner. In agreement with recent experimental data, we show that the model successfully develops latent structures essential for task-solving (cue-dependent "splitters") while excluding irrelevant ones. Finally, our model makes testable predictions concerning the co-dependent interactions between split representations and split behavioral policy during their evolution.

Reinforcement learning algorithms, both artificial and biologically inspired, depend critically on the process described being Markovian—that is, actions and values can be assigned to given states (e.g., a place in the environment), irrespective of history[1–3]. Usually, the algorithm is concerned with learning a good policy (i.e., the strategy of the animal or agent) given an appropriate state space. Typically, in a simple 2D physical environment, an "appropriate state space" consists simply of locations within the environment. However, if an animal or an agent is learning a task which depends on previous history or abstract context not described by the state space (i.e., non-Markovian), simple tabular TD-learning (state value) or Q-learning (state-action value) will fail to find the solution. Therefore, one might also consider the problem of reinforcement learning in the inverse: what are the "appropriate" state representations upon which the policy can be described as Markovian, and how can we learn these representations[4,5]?

As an example of a non-Markovian task, imagine an agent starts at the top left of a 2D grid and traverses the space until it reaches a reward port at the bottom right of the grid. Upon reaching the port, the agent is only rewarded if it has previously traversed a cue location and is punished otherwise (Fig. 1a). The optimal policy cannot be accurately described by single scalar values assigned to states (or state transitions) if they are defined as locations in this 2D space. This can be seen by examining the state-action values for a state immediately before the reward port (state 8 in the example of Fig. 1a)—the transition from state 8 to the terminal state, state 9, is +1 if the agent has visited the cue state, but −1 if it has not. For example, in Q-learning, $Q(8,9)$ (the value for a transition between states 8 and 9), will not converge to the optimal policy. The most basic solution to this problem is to create two copies of that state (8' and 8", in this example) before learning, using the external knowledge that the task depends on two cases; one where the agent has passed through the cue location, and one where it has not. However, this assumption breaks causality from the agent's perspective. We are left with quite a conundrum: the only way to know if two copies of that state are required is to learn the task, but the only way to learn the task is to have two copies of that state (what came first, the chicken or the egg?).

This is one example of a more general problem: real-world tasks often depend on complex combinations of sensory information, internal states, context, etc. which themselves are unknown prior to learning. Agents in these tasks must either pre-assign or learn state representations to create an actionable map of the environment. Animals must similarly represent the state of the environment (or task) via

[1]Department of Bioengineering, Imperial College London, London, UK. ✉e-mail: i.cone@imperial.ac.uk

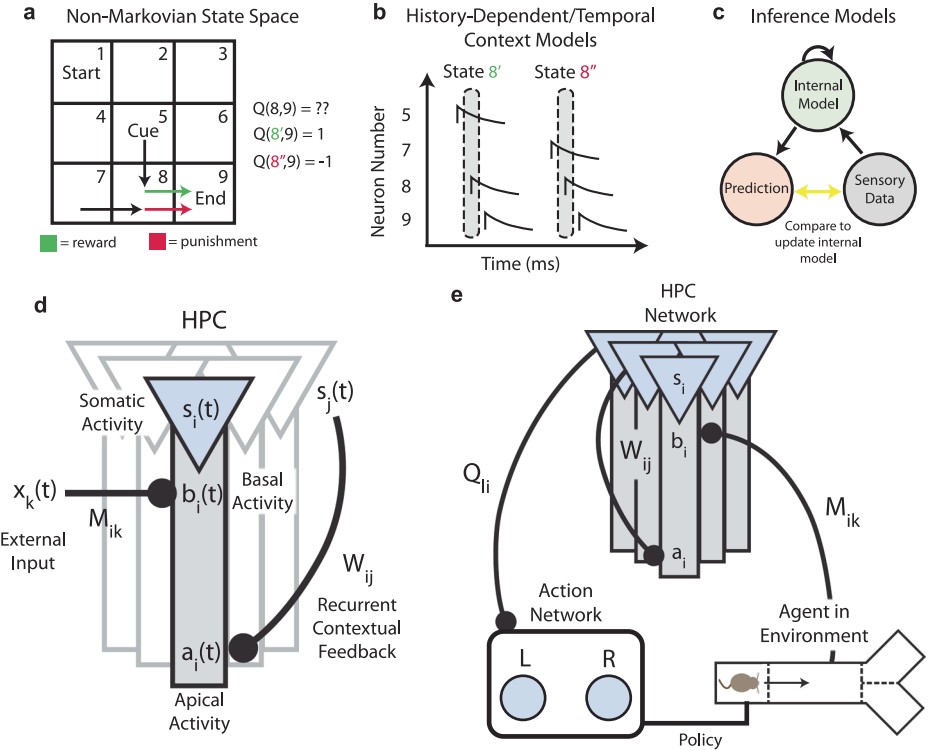

**Fig. 1 | Feedback in three-compartment HPC neurons enables context-sensitive representations. a** A 2D grid environment, in which the agent must visit a cue state (state 5) to receive a reward when it reaches the end state (state 9). Bottom, state transition 8–9 can be either rewarding (if preceded by 5) or punishing (if preceded by 7), leading to ambiguity in the value of the 8–9 transition. **b** Some models include a history-dependence a priori as part of their state representation. Here, the population activity vector is equivalent to the state vector at that time, so the two potential "8" states (shown by the dotted ovals), are disambiguated. **c** Alternatively, inference models compare internal predictions to external sensory observations,

and update their internal models based on errors between their predictions and observations. Successfully trained inference models learn the latent structure of the task as part of their internal model. **d** Schematic of the hippocampal network model we propose. The network receives external inputs $x(t)$ into a basal dendritic compartment $b(t)$. Somatic activation $s(t)$ is a combination of basal activity $b(t)$ and recurrent apical feedback $a(t)$. **e** Schematic of the full model, including action neurons, which receive input from the representations in the hippocampal network and dictate the agent's decisions in its sensory environment, providing a closed loop between the environment and the network.

patterns of neural activity (a "state representation"), which can then act as substrates for planning, behavior, memory, etc. This latent neural structure is often referred to as a "cognitive map", the idea of which has been very closely tied to the observed activity of hippocampus[6–8]. Cells in hippocampus display fields specific to a variety of environment and task variables, including place, time, lap, evidence accumulation, and more[9–15]. These fields can emerge over the course of task-learning and can be induced artificially at the single-cell level[16–18]. If activity in hippocampus is indeed consistent with a cognitive map of state representations, how might animals perform representation learning in order to form these maps?

First, one could imagine a network of neurons which has pre-existing complex representations prior to the task. For example, one might consider using a "temporal buffer" which tracks history in the network, or feeding activity into a liquid state machine, so that all possible sequences create a separate latent state[19] (Fig. 1b). However, reinforcement learning in these overcomplete state spaces is very slow and computationally expensive, scaling exponentially with the number of states (the so-called "curse of dimensionality")[2,3]. A second option is to learn the structure of the environment through prediction and inference[19] (Fig. 1c). Some recent example models of the cognitive map include the Tolman-Eichenbaum Machine (TEM) and the Clone Structured Cognitive Graph (CSCG)[20,21]. The CSCG for example, presumes that each state initially has many copies, and then uses an expectation maximization (EM) algorithm to modify connections between these copies. However, the number of copies chosen for each state is still a parameter chosen a priori by the modeler. Further, plasticity rules

involved in these inference models are generally non-local, rendering them difficult to map onto specific biological processes in hippocampus.

To theorize how hippocampus might form cognitive maps in a biologically plausible manner, we propose a network model which uses local, single-cell plateau-based learning rules to develop population-level cognitive maps, allowing it to learn complex tasks. Our model learns these state spaces and tasks simultaneously in an iterative manner, and cells encoding abstract state variables arise via learning the abstract logic of a given task, rather than existing a priori. Our model's results are compared and validated against recent experimental results, in which the induction of splitter cells was only possible when a representational split was required to solve the task[18]. Finally, our model makes testable predictions about the potential codependence between learned latent hippocampal representations and learned behavior.

## Results

### Compartmentalized feedback in model HPC neurons enables context-sensitive representations

To demonstrate how hippocampus (HPC) might learn task-dependent representations mechanistically, we create a three-stage closed-loop model. The first stage consists of the external environment, which is a Y-maze traversed by our agent in 2D Euclidean space. The second stage is a network of model HPC neurons, which receive both external and recurrent inputs. The activity of the HPC network is projected to a third stage, a set of "action" neurons which dictate the agent's decisions within the external environment. As the agent learns a given task, the

HPC network develops appropriate latent representations upon which the agent can develop a successful policy.

Each neuron in our model HPC network has three compartments —the soma, the apical dendrites, and the basal dendrites (Fig. 1d). The basal dendrites receive input about external sensory information, while the apical dendrites contain feedback from other somata, via a recurrent weight matrix $W_{ij}$ (see "Methods"). The soma receives input from both dendritic compartments, such that the somatic activity $s(t)$ outputs a combination of external information (basal activity, $b(t)$), modulated by the recurrent feedback which encodes the latent structure (apical activity, $a(t)$). The degree to which the somatic activity is dependent on recurrent, apical feedback is determined by a modulatory factor which depends on the sum of the incoming synaptic weights onto the apical dendrites (see "Methods"). In practice, this modulatory factor $\beta$ regulates how much the soma is "listening" to its dendrites. For small $\beta$, the soma is solely a readout of basal activity, while for large $\beta$, the somatic activity is a combination of basal (external) and apical (recurrent) inputs. This function is crucial, since it allows for the soma to express either context-agnostic or context-sensitive representations, depending on the state of the network. While we do not directly model a biophysical process for this compartment-specific modulation, mechanisms of local dendritic inhibition[22], branch strength potentiation[23], and intracellular calcium release[24] have been shown to modify the influence of dendrites on the soma (or the co-tuning between the two).

Learning in the model consists of a plateau-dependent three-factor rule (see "Methods"), in agreement with experimental results which observe the formation of CA1 fields after so-called "plateau potentials"[25–27]. This rule depends on an eligibility trace of pre-synaptic activity $e_{pre}$, post-synaptic activity $s_{post}$, and reward $r$ above a baseline rate $r_0$. The two coincident factors (pre- and post-synaptic activity) are evaluated at the occurrence of a plateau potential, i.e., at time $t_{plateau}$. The recurrent weights $W_{ij}$ are updated in batch at the end of the trial. The recurrent weights $W_{ij}$ determine the modulatory recurrent feedback received in the apical compartment $a(t)$. Crucially, this three-factor rule allows the network to reorganize sequentially activated pairs of state representations that lead to reward into a new state. In other words, this allows first-order representation $s_{first} = x_i$ (e.g., place only) to be combined into second order representation $s_{second} = x_i x_j$ (e.g., place and cue context). Higher order representations can then be learned as combinations of first and second order representations, etc. Simpler learning rules which only consider one state and its connection to reward (i.e., TD) seem therefore incapable of learning these higher-order combinations, and more complicated rules which can learn higher order relations such as backpropagation within a recurrent neural network are typically non-local and therefore not typically biologically plausible[28].

Attached to our hippocampal network is a network of two "action" neurons (representing the two possible turn directions, left or right) which dictate the agent's turning decision in the environment. These action neurons $v(t)$ are connected to somatic activity $s(t)$ through weights $\mathbf{Q}$, which can be broadly interpreted as state-action values. The information in the model thereby follows a closed loop: The agent moves in the environment at a constant velocity $dx/dt$ and observed external stimuli $x(t)$ are fed into our representation layer as basal activity $b(t)$. The basal activity $b(t)$ combines with recurrent feedback $a(t)$ to produce internal representational states $s(t)$, which then activate action neurons $v(t)$ through weight matrix $\mathbf{Q}$. As the agent approaches the choice point, these action neurons will dictate it to turn right or left. After turning, the agent reaches one of the end branches of the track, and reward is either presented or omitted (the conditions of which depend on the task). Finally, the agent is teleported back to the beginning of the track for a new trial, and the process repeats (Fig. 1e).

## Network learns task-dependent latent representations and does not integrate task-irrelevant representations

To test our model's ability to learn task-dependent latent representations, we place an agent in a Y-maze environment. This is to mimic a recent experiment which reported the emergence of hippocampal "splitter" cells, or cells which fire in a given location only in a given context[18]. The agent is presented with one of two possible visual cues, A or B (represented here as red vertical bars or black horizontal bars), before walking along a track (C). Upon reaching the end of the track, the agent can turn left or right (to locations D or E) to potentially receive a reward (Fig. 2a). We train the agent on one of two possible tasks. In both tasks, we use the same artificial induction protocol, wherein half of neurons receive location-specific plateaus after presentation of A, and the other half of neurons receive location-specific plateaus after presentation of B. The induction protocol is applied for each trial during learning, so for a single trial, half of the population (either the "B-type" or "A-type", depending on which of the cues is shown) receives a location-specific plateau (Fig. 2b). For the first task, reward was contingent on the initial cue which was presented to the agent, such that if the agent saw cue A, the reward would be in port D with 100% certainty, and if the agent saw cue B, the reward would be in port E with 100% certainty (cue-dependent task, Fig. 2c). In another task, reward was randomly given, such that D or E had a 50% probability of containing reward, regardless of the visual cue shown (random reward task, Fig. 2d).

We observe that, in the cue-dependent task, neurons in our model HPC develop "split" representations in agreement with the logic of the induction protocol. That is, cells which were artificially injected with current at a given cue-location combination during training develop fields that are selective to that cue-location combination after training (Fig. 2e,i). An example cell (Fig. 2e,ii) only fires at position C following presentation of the A but has no firing field at any location after the presentation of B. The development of cue-dependent splitter cells we observe in the cue-dependent task is in agreement with recent experimental findings of task-dependent hippocampal representations[9,15,18,29–32] (Fig. 2e,iii). However, if we train the agent on the random reward task, we observe that even when we induce plasticity only for a given cue-location pair, the resulting fields do not retain this information, instead becoming generic "place" fields (Fig. 2e,iv). The same example cell which had, in the cue-dependent task, developed a conjunctive field (location C if preceded by A) in line with the induction protocol, in this case encodes generally for location C regardless of the preceding cue (Fig. 2e,v). Experimentally too, a cue-dependent field could not be induced in a single cell when the animal was trained on the random task[18] (Fig. 2e,vi).

The combination of these results shows that the artificial split induction protocol (Fig. 2b) is necessary but not sufficient to form splitters. Instead, the logic of the induction protocol is only integrated into the network if it leads to an underlying representation which improves the behavioral acquisition of reward. Further, since the agent in our simulations starts with a naïve policy (random action selection), the knowledge of whether a given representation will lead to increased rewards is unknown a priori (leading again to our chicken and egg problem; one needs splitters to learn the policy, but one needs to learn the policy to know whether to integrate splitters). The combination of these factors makes learning non-trivial, despite the fact that the split induction protocol itself ostensibly contains the split logic. Crucially, in order to match both experimental results (Fig. 2e,iii,vi), our model must assume that memory of the cues is in fact inaccessible later on in the track, prior to learning (otherwise, we would develop splitters in both the cue-dependent and random tasks). Instead, in order to form splitters only in the cue-dependent task, the network simultaneously must (1) learn to propagate the memory of the cue throughout the track, and (2) learn that the split state representation of cue memory and current location is beneficial for behavioral outcomes. Note that

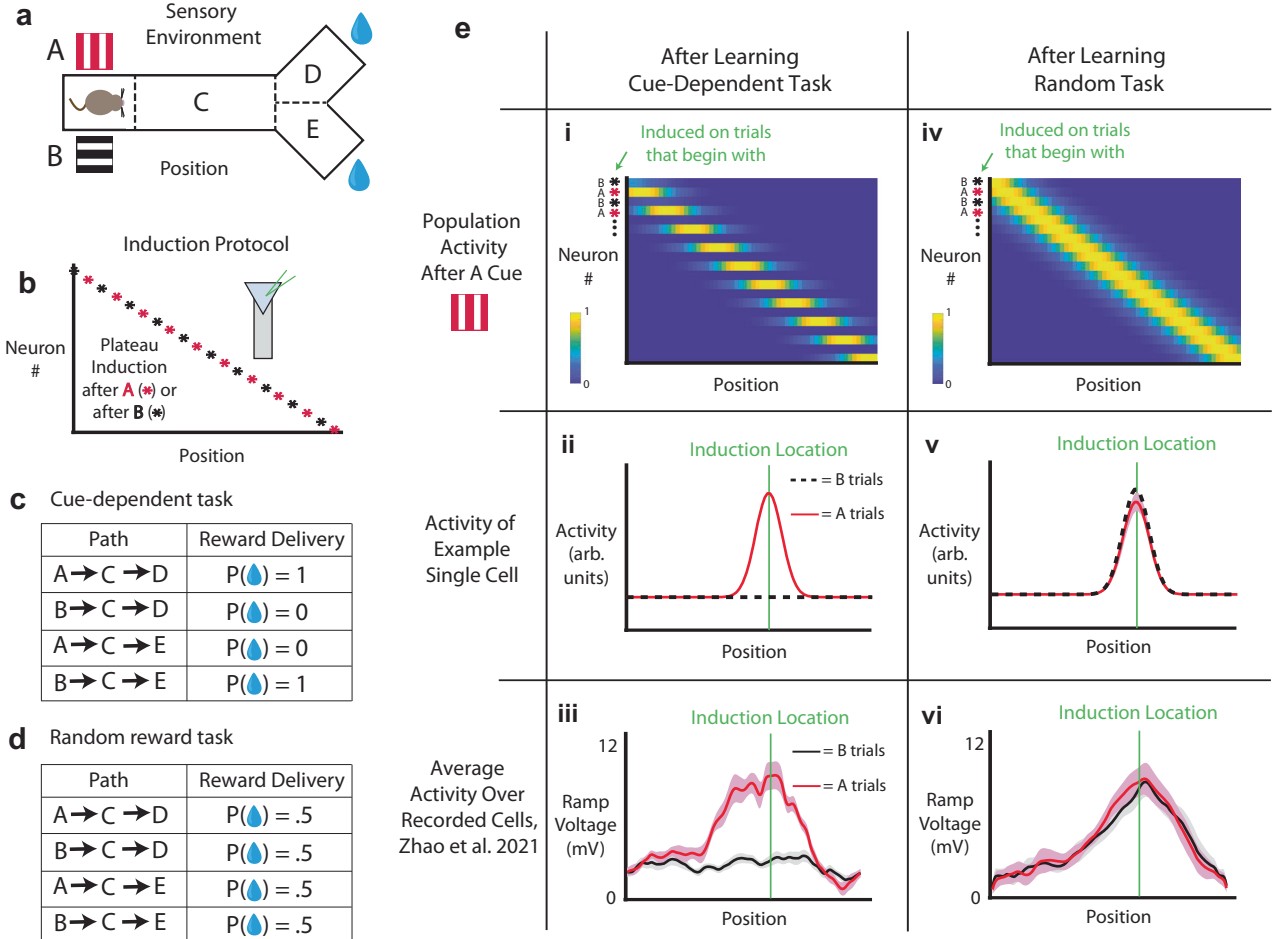

**Fig. 2 | Splitters emerge when dictated by task structure. a** Task environment for simulation (to mimic the experimental design of Zhao et al.[18]). The agent is presented with one of two possible cues, A or B (here represented as red vertical bars or black horizontal bars, respectively), before advancing through the track (C). It can then choose to go to either branch D or E to potentially receive a reward. **b** Fields are induced in model HPC cells via the induction of a "plateau potential" at a certain location after a certain cue. Half of the neurons are induced on trials which begin with cue A, and half are induced on trials which begin with cue B. **c** The cue-dependent paradigm, where location D contains reward if preceded by cue A, and location E contains reward if preceded by cue B. **d** The random reward paradigm, where locations D and E each result in a 50% chance of reward delivery, regardless of the cue shown. **e** Simulated population activity (i, iv), single-cell activity (ii, v), and experimentally recorded activity after learning (data from Zhao et al.[18]) (iii, vi), for both the cue-dependent task (i–iii) and the random reward task (iv–vi). Notably, the same induction protocol generates splitters in the cue-dependent case, and non-specific place fields in the random reward case. Red, trials which began with an A cue, black trials which began with a B cue. Shaded area represents SEM. $n = 20$ neurons for simulated activity and $n = 12$ cells for experimental data[18].

since splitting the state representation and/or changing the policy cannot improve average outcomes in the random reward task (average reward will be 0.5), our representations remain as generic place fields in that case.

**Behavior co-evolves with internal representation**

From the simulations shown above, we can infer the state representations and behavior before and after learning (Fig. 3a). However, since both internal state representations and behavior are plastic, we can also examine their dynamic evolution during learning. We quantify learned behavior by measuring a running average of the fraction of correct turns the agent makes during training on the cue-dependent or random task. To quantify the state representation, we introduce a measure of "splitness" for neurons in our HPC network (firing rate (neural activity) on A cue trials−firing rate (neural activity) on B cue trials). Since the task requires split representations to be solved, this can be understood as a sort of fitness of the state space to the task.

From the perspective of reinforcement learning, learning of appropriate actions is contingent on an accurate state space, so one might expect the evolution of behavior in our model to lag the evolution of state representations ($\tau_{behavior} > \tau_{state}$). However, here

representation (Fig. 3b, top) and behavior (Fig. 3b, bottom) evolve together, ($\tau_{state} \approx \tau_{behavior}$) because (1) we cannot learn the split behavior without splitters, and (2) we also cannot learn the splitters without split behavior. Our model is able to break this loop owing to stochasticity in behavior, which acts as a symmetry-breaker to the underlying representation. Our three-factor learning rule reinforces breaks along dimensions useful to the task, while breaks along null dimensions will relax back to zero. Improper policy (i.e., going to the wrong reward port) will degrade the state representation, as unrewarded state pairs are depressed. Meanwhile, proper policy reinforces state pairs which lead to reward. Therefore, it is key in our model that state representations iteratively improve behavioral performance, and vice versa, in a cooperative and codependent (rather than merely concurrent) manner. These results are in agreement with experiments which observe task-relevant hippocampal representations arising on a similar timescale as the relevant behavior[9,29,30,32,33]. This idea acts in stark contrast to reservoir type models[34,35], where an output has access to a pool of inputs which form an overcomplete basis set, and few-shot learning can quickly select any desired representation of the inputs (such as cue memory and current location), without needing to train beforehand on any task. Instead, our simulations suggest that both task and

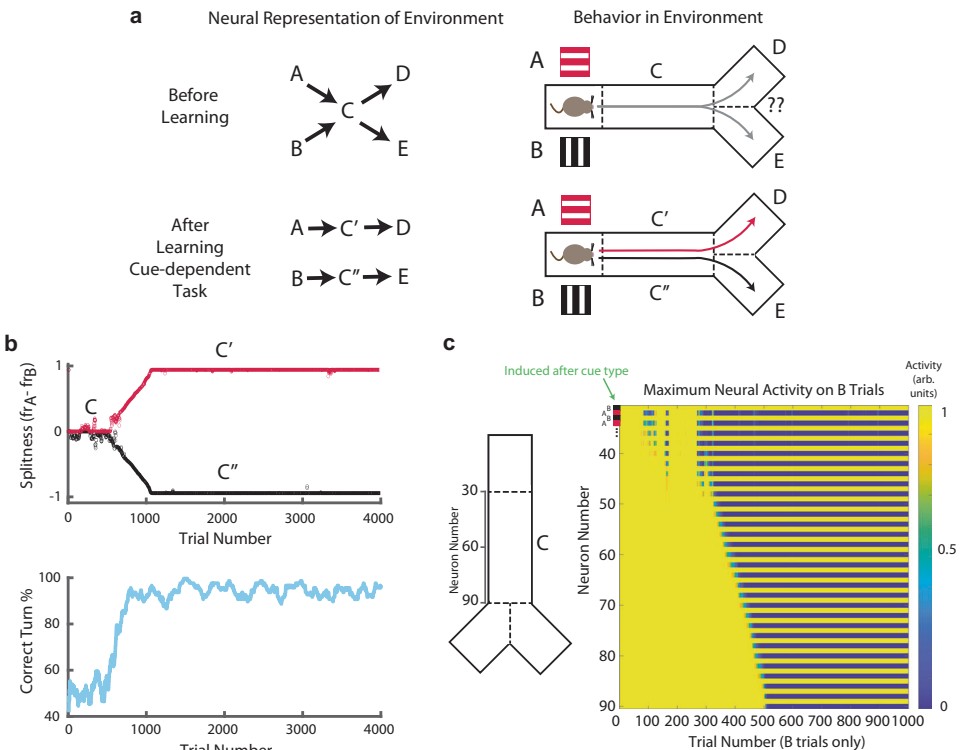

**Fig. 3 | Behavior and representation iteratively improve each other. a** Schematic of representation (left) and agent behavior (right), before (top) and after (bottom) learning the cue-dependent task. Initially, the agent cannot implement the optimal policy at C since a single state cannot incorporate two separate state-action values. After training, the state has been split, and the policy has been learned. **b** Top, difference in mean firing rates (activity) on A and B cue trials ("splitness") of the two populations over the course of learning. Activity is plotted as a running mean over 100 trials. Bottom, behavioral performance, shown as the percentage of correct turns over the course of learning. Behavioral performance is plotted as a running mean over 100 trials. **c** Left inset, neuron index corresponding to a given location in the track. Right, evolution of population activity on B trials over the course of training. Odd numbered neurons are induced on B trials, while even numbered neurons are induced on A trials. Splitters emerge in a "zipper"-like fashion, starting from the part of C closest to the cue zone (neuron 30), and propagating along until the end of the track nearest to the reward (neuron 90). This zippering creates two feed-forward "chains" of splitters, one propagating the memory of cue A, and the other propagating the memory of cue B.

representation must be learned simultaneously, as opposed to complex task representations all existing, or being accessible, a priori.

Examining the evolution of the population activity (Fig. 3c), we see that for the first ~200–300 trials, the network is comprised mostly of place fields, with splitters stochastically appearing and disappearing near the start of the track due to spurious asymmetries in the agent's behavior (e.g., a few actions in a row which happen to coincide with the optimal policy). Eventually, these stochastic fluctuations create a bias (in representation and behavior) that overcomes the exploration noise in our action network. Then, over the course of the next couple of hundred trials (~300–500), the population slowly splits, from the start of the "C" component of the track (neuron 30) to the end of the "C" component of the track, in a "zipper-like" fashion. This "zippering" process acts to create two feed-forward sequences of neural activity—one related to cue A (C′), and one related to cue B (C″). It is through these learned sequences that the memories of cue A and cue B propagate throughout the delay period, thereby enabling post-synaptic plateaus to select for this contextual information even long after the cues have ended (Supplementary Fig. 1). This evolution of population activity mirrors an evolution in the recurrent weights (Supplementary Fig. 2), whereby neurons that represent positions later in the track slowly become more interconnected with same cue-type neurons from earlier in the sequence. Owing to this phenomenon, our model predicts that the split representation emerges first in cells which encode locations nearest to the cue.

### Representation and behavioral policy are co-dependent

While we have demonstrated that behavior and representation evolve cooperatively during training, we can also examine their codependence after training, by perturbing behavior and observing the changes in representation, and vice versa.

To test behavior's effect on representation, we performed a perturbation simulation where we reset behavior to chance once the agent has learned the task (Fig. 4a). We find that the state representation in our model re-merges, since neither C′ nor C″ result in positive reward above baseline (Fig. 4b, c). Note that this representation is equivalent to the one learned in the random reward task (again, neither C′ or C″ result in positive reward above baseline). The ability of behavior to modulate hippocampal representations is supported by recent experimental results which show that the quality of cognitive maps in human subjects is experience dependent[36]. Notably, the collapse of the split representation after a behavioral reset is incongruent with models of representation that rely on a pre-existing, general memory (such as a reservoir) that exists prior to (or absent of) behavior. Instead, it suggests that complex representations are dependent on the selective, learned propagation of specific memories. In the splitter task, this means that splitters cannot exist at the end of the track exclusively, as they are reliant on the contiguous memory of the cue. This memory is propagated by the network through the previous splitter cells in the sequence (the existence of which depends on their ability to improve behavior). Altogether, our model predicts that the split representation (emerging from training on the cue-dependent task) would collapse after (1) switching to the random task, or (2) switching to random behavior (perhaps through artificially randomizing whether the animal ends in the left or right "location" in a VR environment).

We also perform the opposite manipulation where we artificially induce a bias in the network's latent representation and examine its

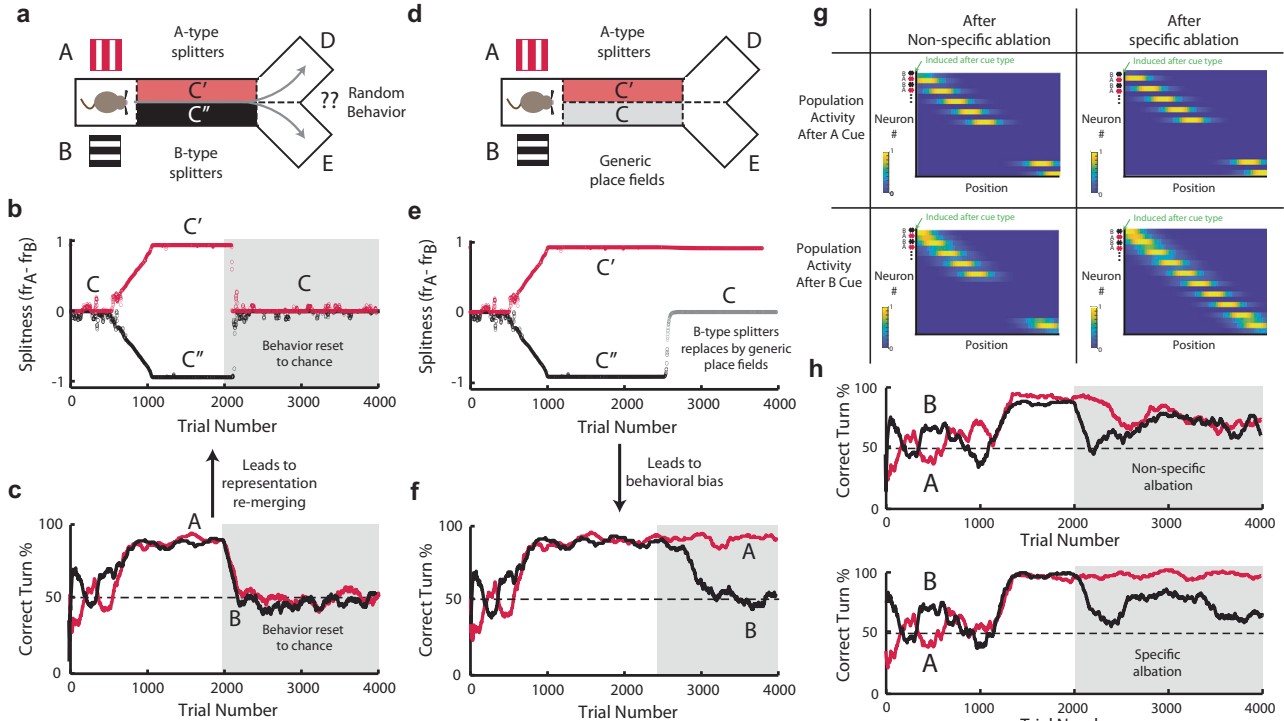

**Fig. 4 | Biases in representation lead to biases in policy. a** After successful learning, behavior is forced back to chance. **b** Difference in firing rates on A and B trials ("splitness") of two example neurons over 4000 trials of learning. Representation re-merges since neither C' nor C' results in positive reward above expectation. **c** Behavior, plotted as the fraction of correct turns on A trials (red) and B trials (black) over 4000 trials of learning. Left/right turn behavior is fixed to chance at trial 2000, chance behavior indicated by gray boxed area. **d** After successful learning, B-type splitters are replaced by generic place fields. **e**, **f** Same as in (**b**, **c**), but for the case of forced representational bias (**d**) leading to behavioral bias

effects on behavior (Fig. 4d). To do so, after initially training on the cue-dependent task, we selectively prune the weights which project to the B-dependent splitters. This produces two functional populations: A-dependent splitters (C') and generic place cells (C) (Fig. 4e). We then examine how behavior co-evolves with this biased representation. We observe that on A trials, the agent learns to always turn to D (correct policy), while on B trials, the agents on average behave at chance (Fig. 4f). This occurs even though no restrictions are placed on behavioral learning. Instead, since the latent representation lacks B-specific splitters, the agent cannot assign unique state-action values in the B trials, and therefore fails to find the correct policy following the B cue (i.e., the model displays a feature-specific behavioral deficit). Experiments have also found that task-relevant representations are more stable than generic fields[29,30,33], which may underlie their importance in supporting and maintaining task-performance.

Experimental studies have also shown that errors in single-trial behavioral performance are correlated with a decrease in task-specific representations in hippocampus during that trial[37–39]. To examine this effect, we test whether we can induce long-lasting biases in the policy (continued errors of a specific type over many trials) by artificially modifying the underlying representation. To do so, we ablate splitter cells after initial training (Fig. 4g). Here we consider two types of ablations in our network—general and specific. For the general ablation, we inactivate a fraction of splitters in the network after behavior reaches asymptote, regardless of their corresponding A- or B-type identity. Once the splitters have been removed, the agent's performance drops in both the A- and B-type trials (Fig. 4h). However, if we perform an ablation which only targets splitters of a specific identity

(f). **g** Population activity after general (left) or specific (right) ablations of splitters. Top row shows network activity following presentation of an A cue, while bottom row shows network activity following presentation of a B cue. **h** Behavior, plotted as the fraction of correct turns on A trials (red) and B trials (black) over 4000 trials of learning. Top panel, general ablation is performed at trial 2000, and turn performance is impacted across both trial types. Bottom panel, B-type splitters are specifically ablated at trial 2000, and turn performance is only impacted in B-type trials. For all plots, curves of activity and behavior are plotted as a running mean over 100 trials.

(in this case a fraction of B-type splitters are inactivated), the agent demonstrates a feature-specific behavioral deficit which lasts across many trials, as it is unable to reach behavioral asymptote specifically on the B-type trials (Fig. 4g, h). Notably, ablation of some splitters can also degrade the surrounding representation, as cells within the population are themselves responsible for the propagation of the cue memory.

## Slow, population-level integration of task structure enables fast single-cell learning

The timescales of learning (hundreds or thousands of trials) which we have so far shown far exceed the known timescales of the induction of splitters and place cells in hippocampus. Experimentally, plateau potentials have been shown to generate place fields or splitters after only a handful (<10) of trials[25–27]. However, in our model we are training the network and the agent's behavior from scratch. The iterative nature of the cooperation between behavioral and representation learning (i.e., rewarded behaviors improve representations, which thereby increase rewarded behaviors, etc.) is a slow process that dominates the overall timescale of learning in the network. However, this lengthy process of building the population level "cognitive map" then might allow the network to quickly integrate new single units. To test this, we hold out two neurons from the initial phase of learning (one an A-type splitter and one a B-type splitter), keeping their inputs at zero. After training, we stimulate the external inputs to induce the cell, allowing for their weights to undergo plasticity (Neurons 80 and 81 in Fig. 5a). During the initial training, as we have previously shown, cells take hundreds of trials to form splitters, with the majority of cells reaching the split criterion during the "zipper" phase occurring from trial

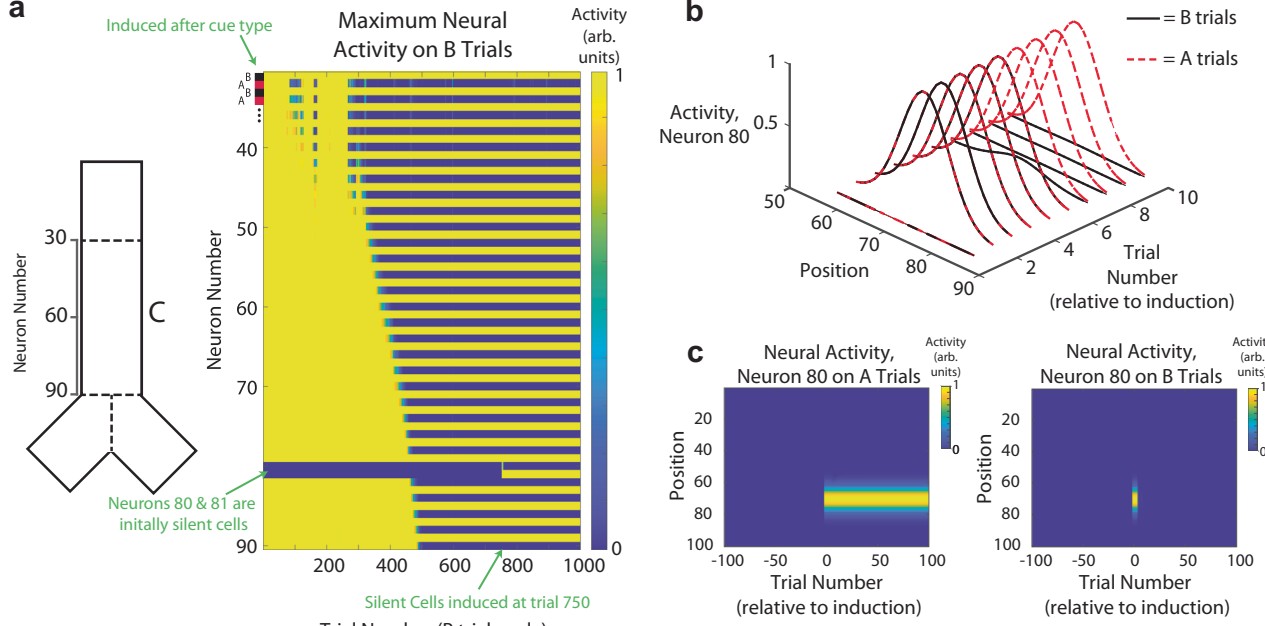

**Fig. 5 | Integration of task structure allows for few-shot learning of context-specific cells. a** Left inset, neuron index corresponding to a given location in the track. Right, evolution of population activity on B trials over the course of training, with neurons 80 and 81 held out of learning until trial 750. After the split population has emerged, plateaus are induced in the two held out neurons, and they are quickly integrated into the split population. **b** Activity of a held-out silent cell

(neuron 80), on both A and B trials, relative to the trial of initial induction after behavioral learning. After only six trials, the cell has developed a split representation of the A and B trials, only demonstrating its location-specific field after the presentation of cue A. **c** Rate map of the one of the held-out neurons (neuron 80), centered around its induction trial after behavioral learning (trial 750 in **a**). After induction, the neuron becomes an A-type splitter.

~300–500. However, upon being induced, the holdout neurons initially form generic place fields, before quickly integrating themselves into the latent A- and B-cue sequential structure (neuron 80 shown in Fig. 5b). This integration into the split representation now only takes ~10 trials (six in the case of neuron 80), in agreement with the timescales of single-cell plasticity observed experimentally (Fig. 5c)[25–27]. Since the initial evolution of splitters is slow and codependent with behavior, our model predicts that the few-shot nature of learning splitters (in particular, as opposed to generic place fields) via artificial induction of BTSP is only applicable after, or during the late stages of behavioral learning.

The integration of the task also allows for the network to generalize across environments which map onto the same latent structure. To demonstrate this, we allow plasticity in the input (alongside recurrent and behavioral learning) in the cue-dependent task and introduce a novel cue and novel reward port after initial training. We find that the network tracks changes in the input by relabeling or remapping the existing internal representations onto the new external stimuli. Though the inclusion of plasticity in the input increases variance in the representation, it does so without loss of behavioral function, and both the representation and behavior are unaffected by the switch to the novel components of the environment (Supplementary Fig. 3).

## Discussion

The hippocampus has long been thought to operate as a "cognitive map", but the process by which it forms these maps is still unknown[6–8]. A principal difficulty in building cognitive maps is learning the appropriate state representation for tasks or environments which are described by complex, higher-order relations. Various models have been introduced to learn artificial state spaces (state discovery/representation learning)[40–42], but our model attempts to directly link the emergence of population-wide cognitive maps to observed single-cell, plateau-based plasticity mechanisms in hippocampus. Further,

our model replicates recent experimental results which show CA1 cells forming complex fields only when they are relevant to a behavioral task[18]. We observe in our model that behavior and representation iteratively improve each other, and that controlled modifications to either can lead to changes in the other. This lockstep evolution of behavior and representation leads to a characteristic "zippering" phenomenon observable in the representation during learning. Our model predicts that (1) the split representation will emerge in cells which encode locations nearest to the cue, (2) artificially randomizing the behavior of the animal (e.g., via manipulating a VR environment) should "merge" a learned split representation, and (3) few-shot learning of individual splitter cells via BTSP induction will only be successful after (or during the late stages of) behavioral learning of a split task.

Our model network learns to behave by learning the task structure, rather than by building a general map of an environment. While it is true that hippocampal representations emerge in environments without an explicit task structure[12,13], and common representations (such as physical space) are reasonable for an agent to generalize and learn in an unsupervised manner, unsupervised learning of higher-order state environments quickly becomes untenable, as the number of potential higher-order states grows exponentially with the size of the first-order state space (curse of dimensionality)[2,3]. As such, it is likely useful to learn only those higher-order, complex representations which are necessary to maximize reinforced or self-supervised objectives. Regardless, evidence shows that complex representations in hippocampus are either (1) more likely to, or (2) exclusively emerge in tasks which require them[9,15,18,29,31]. While in this work we interpret the modulated learning term ($r$) as explicit reward, this term could represent intrinsic reward or some variables related to an internal statistical model.

Our model considers fixed plasticity protocols, i.e., we as an outside observer choose to induce splitters in this task. While we do show that induction protocols unnecessary for the task are not integrated into the network, our network does not spontaneously generate

**Table 1 | Model parameters**

| Parameter | Value | Units | Description |
|---|---|---|---|
| $N_{inp}$ | 120 | – | Number of neurons in external network |
| $N$ | 120 | – | Number of neurons in hippocampal network |
| $N_{act}$ | 2 | – | Number of neurons in action network |
| $dx/dt$ | 1 | dv (arb.) | Velocity of agent in maze |
| $L$ | 100 | dx (arb.) | Length of maze |
| $\sigma$ | 5 | dx (arb.) | Standard deviation of input Gaussians |
| $\tau_e, \tau_{ea}$ | 10, 20 | dt (arb.) | Time constants of eligibility traces |
| $\tau_a, \tau_v$ | 40, 10 | dt (arb.) | Time constants of apical compartment, action neurons |
| $\sigma_v$ | N(0,0.75) | – | Noise, action neuron activity |
| $\sigma_p$ | N(0,5) | – | Noise, policy selection |
| $m_a, m_\beta$ | 1, 5 | – | Stretch coefficient, activation function |
| $c_a, c_\beta$ | 5, 1.4 | – | Offset, activation function |
| $t_{choice}$ | 60 | dt (arb.) | Time of turn choice |
| $p_{rand}$ | 10 | % | Chance of random turn |
| $n_{trials}$ | 4000 | – | Number of trials |
| $r_0, r_q$ | 0.5, 0.6 | – | Reward expectation |
| $t^i_{plateau}$ | i | dt (arb.) | Time of induced plateau for neuron $i$ |
| $M_{ik}$ | $\delta_{ik}$, U(0,1e$^{-4}$) (S2 only) | – | Weight matrix, input layer to representation layer. Identity matrix (fixed) for all but Supplementary Fig. 2) |
| $W_{ij}$ | U(0,1e$^{-4}$) | – | Initial values, recurrent weight matrix, representation layer (before learning). Uniform distribution between 0 and 1e$^{-4}$ |
| $Q_{li}$ | U(0,1e$^{-4}$) | – | Initial values, weight matrix, representation layer to action layer (before learning). All initial values the same |
| $I_{ml}$ | $\begin{pmatrix} 0 & -0.125 \\ -0.125 & 0 \end{pmatrix}$ | – | Recurrent weight matrix, action layer (fixed) |
| $M_{max}$ | 0.75 | – | Upper bound for input weight (Supplementary Fig. 2 only) |
| $W_{max}$ | 0.15 | – | Upper bound, recurrent weights |
| $Q_{min}, Q_{max}$ | −0.15, 0.15 | – | Lower and upper bounds, action weights |
| $\eta^W, \eta^Q, \eta^M$ | 0.0006, 0.0003, 0.15 | – | Learning rates, recurrent weights, state-action weights and input weights (input learning for Supplementary Fig. 2 only) |
| $\lambda_w$ | 0.025 | – | Decay constant, recurrent weights |

plateaus or meta-learn the induction protocol. In natural formation of fields in hippocampus, of course, this loop would also be closed, and likely depends on other areas such as the entorhinal cortex[43]. Though random induction alone is insufficient (Supplementary Fig. 4), one can imagine that through a noisy process, the network might sample potential latent states via one-shot learning to create quick combinations of external inputs/recurrent feedback. If these latent states are useful, perhaps they are reinforced and remain in the state space, remapping otherwise. It remains to be seen how a biophysically plausible model network could learn appropriate plateau induction protocols, task-relevant representations, and policies simultaneously. Additionally, our model does not consider other forms of plasticity aside from BTSP which likely play an important role during field formation in hippocampus[16]. Likely, the few-shot plasticity of BTSP is combined with more conventional, slower forms of plasticity (Hebbian learning, homeostatic mechanisms, etc.) in the formation and maintenance of these cognitive maps.

Our model hippocampal network presented here demonstrates that single-cell plateau-based learning can interact with behavioral learning to generate a population level cognitive map via cooperative improvement of behavior and representation. This method avoids problematic a priori assumptions of the structure of a given state space and presents a potential pathway for further research into the online, in vivo formation of task-relevant hippocampal cognitive maps.

## Methods
### External input
All parameters for the following methods are included in Table 1. The external sensory environment is modeled in the form of stereotypical

1-D positional tuning curves of the following form:

$$u_k(x) = e^{-\left(\frac{x-x_k}{\sigma}\right)^2} \tag{1}$$

where k indexes over K total inputs, and $x_k$ are the locations of the tuning curve centers, which have standard deviation $\sigma$. For simplicity, we model the animal as running at constant unit velocity through the track, such that $x = t$. HPC neuron $i$ receives input $M_{ik}u_k(t)$. $M_{ik}$ is equivalent to the identity matrix in this work, though in general, one might consider these weights to be plastic, as we do in Supplementary Fig. 3.

### HPC network
The rate activity, $s_i(t)$, of each HPC neuron $i$ is described by the following three-compartment model:

$$s_i(t) = (1-\beta_i)b_i(t) + \beta_i b_i(t)\gamma_a[a_i(t)] \tag{2}$$

$$\beta_i = \gamma_\beta\left[\sum_j W_{ij}\right] \tag{3}$$

$$\gamma_f(x) = \frac{1}{2}\left[\tanh\left(m_f\left(x-c_f\right)\right)+1\right] \tag{4}$$

where $f = a$ for the apical non-linearity and $f = \beta$ for the non-linearity on the incoming weights. $m_f$ and $c_f$ are constants.

$$b_i(t) = M_{ik}u_k(t) \tag{5}$$

$$\tau_a \frac{da_i}{dt} = -a_i + W_{ij}s_j(t) \tag{6}$$

The neuron receives external input $u_k(t)$ into its basal compartment via weights $M_{ik}$, and receives recurrent feedback into its apical compartment via weights $W_{ij}$ with a time constant $\tau_a$. The somatic activity is comprised of two components: basal activity $b_i(t)$ and basal-apical product $b_i(t)\gamma_a[a_i(t)]$. The degree to which the somatic activity is influenced by each component is modulated via a non-linear sum of weights $\beta_i$. In short, when the incoming weights to the apical compartments is large, the somatic activity is approximated by $b_i(t)\gamma_a[a_i(t)]$, while when the incoming apical weights are small, the somatic activity is essentially $b_i(t)$. While we do not directly model a biophysical process for this compartment-specific modulation, mechanisms of local dendritic inhibition[22], branch strength potentiation[23], and intracellular calcium release[24] have been shown to modify the influence of dendrites on the soma (or the co-tuning between the two). Recurrent weights $W_{ij}$ are modeled here as a single matrix connecting HPC somata to apical dendrites. While we do not directly map our model onto a particular area of HPC, recurrent loops CA1-EC-CA1 or CA3-CA3 could be considered as candidates, as context-sensitive cells have been reported in both CA1 and CA3[17,18]. Apical activity and incoming weights are passed through a non-linear activation function $\gamma$ in (2) and (3).

Somatic activity $s_i(t)$ produces eligibility traces $e_i(t)$:

$$\frac{de_i}{dt} = \frac{-e_i(t)}{\tau_e} + s_i(t) \tag{7}$$

A history of activity for each neuron, $e_i$ is calculated as an exponential filter of the activity, with a time constant $\tau_e$.

Recurrent weights $W_{ij}$ are learned via a three-factor rule which depends on post-synaptic plateau $\varphi[s_i(t)]$, pre-synaptic eligibility trace $e_j(t)$, and reward $r$ above baseline $r_0$:

$$\Delta W_{ij} = \eta^{\boldsymbol{W}}[r - r_0] \int_0^T \varphi[s_i(t)]e_j(t)dt - \lambda_w W_{ij} \tag{8}$$

The total time of the single trial is $T$. Weights are updated in batch at the end of each trial. $\eta^{\boldsymbol{W}}$ is the learning rate for the recurrent weights, and recurrent weights are bounded at $W_{max}$. The plateau occurs upon manual induction and acts as a pass-filter to the learning rule:

$$\varphi[s_i(t)] = \begin{cases} s_i(t), & i = i_{plateau} \\ 0, & i \neq i_{plateau} \end{cases} \tag{9}$$

On unrewarded trials, weights of visited state pairs $\{i,j\}$ will decay as $\varphi[s_i(t)]e_j(t)r_0$. However, for unvisited state pairs this term is zero, so weight decay $-\lambda_w W_{ij}$ is also included in the rule. Weights are updated in batch at the end of a given trial. The integral in (8) can be considered as the accumulation of "proto-weights", which are updated once reward is either delivered or omitted. For main results, the network is trained for 4000 trials.

### Action network

The main network is connected to an action network which determines the real space policy of the agent at the choice point. The activity $v_l$ of units in the action network are described by the following:

$$\tau_v \frac{dv_l}{dt} = -v_l + Q_{li}s_i(t) - I_{lm}v_m(t) + \sigma_v \tag{10}$$

Where $l$ indexes over the two possible choices for turn direction, left (L) and right (R). $Q_{li}$ are the weights connecting the representation network to the action network, and $I_{lm}$ provides mutual inhibition so that the action network has winner-take-all dynamics. Gaussian noise $\sigma_v$ is introduced into the network, and the neurons operate with time

constant $\tau_v$. The action network is queried at $t_{choice}$ for the agent's turn selection, which is decided by $\max\left(\vec{v} + \sigma_p\right)$ (where $\sigma_p$ is policy noise), which results in the agent taking a turn either left or right into one of the reward ports. For a fraction of trials $p_{rand}$, the animal makes a random turn and does not query the action network. After action selection, the selected neuron is fixed at 1 for the next ten timesteps.

Action weights $Q_{li}$ are learned via a three-factor rule, similar to that for $W_{ij}$:

$$\Delta Q_{li} = \eta^{\boldsymbol{Q}}[r - r_q] \int_0^T v_l(t)e_{action,i}(t)dt \tag{11}$$

The three factors are the eligibility trace of pre-synaptic activity for action neurons $e_{action,i}(t)$, post-synaptic activity $v_l(t)$, and reward $r$ above baseline $r_q$. $e_{action,i}(t)$ follows the dynamics of (7), but with time constant $\tau_{ea}$. The total time of the single trial is $T$. Weights are updated in batch at the end of each trial. Though they do not explicitly match the definition of $Q$-values, they are effectively state-action values and thus can be interpreted similarly. Action weights are bounded at $Q_{min}$ and $Q_{max}$. For Supplementary Fig. 3, input weights are learned via the coactivation of pre-synaptic input activity $u_k(t)$ and post-synaptic plateaus $\varphi[s_i(t)]$:

$$\Delta M_{ik} = \eta^{\boldsymbol{M}} \int_0^T u_k(t)\varphi[s_i(t)]dt \tag{12}$$

### Reporting summary

Further information on research design is available in the Nature Portfolio Reporting Summary linked to this article.

### Data availability

All data supporting the findings of this study are available within the paper and its accompanying custom MATLAB code (version 2021a).

### Code availability

All simulations were run via custom code in MATLAB 2021a. The code is available at https://github.com/ianconehed/latent_reps_hpc[44].

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

## Acknowledgements
This work was supported by BBSRC (BB/N013956/1), Wellcome Trust (200790/Z/16/Z), the Simons Foundation (564408) and EPSRC(EP/R035806/1) (C.C.). We are grateful to Toshitake Asabuki and Albert Albesa-González for their comments and feedback on this manuscript.

## Author contributions
I.C. and C.C. conceived and designed the model. I.C. developed and performed the simulations. I.C. and C.C. wrote the manuscript.

## Competing interests
The authors declare no competing interests.
