## [Peer Review File · Nature Communications]

Latent Representations in Hippocampal Network Model Co-Evolve with Behavioral Exploration of Task StructureREVIEWER COMMENTS

Reviewer #1 (Remarks to the Author):

“Latent Representations in Hippocampal Network Model Co-Evolve with Behavioral Exploration of Task Structure”

Animals must pair actions to task structure, and the hippocampus has receptive fields related to these actions. But the link between biophysical plasticity mechanisms and the development of task-relevant maps is not clear. The authors construct a computational model with a hippocampal network and an action network and show that the network develops latent structures that are relevant to solving the task, in agreement with experimental data on splitter cells.

This work is important because it provides a framework to understand how agents might solve non-Markovian tasks, i.e. tasks in which actions depend on more than just the immediate prior state. This of course has applications in neuroscience but will be of interest to any field involving action selection and agent policy construction. The model is well-described, is grounded by experimental observations, and makes experimentally testable predictions.

The alternative models described are modern, and I am not aware of other modeling studies which aim to directly address the question of learning the appropriate state combinations as this manuscript does.

Overall, I am in support of this paper. I would like to see some more comparisons to experimental data as well as justifications for some of the model design choices

Major Points

1. Figure 3 demonstrates that several hundred trials are needed before splitter cells emerge. This seems at odds with the experimental data presented in Zhao et al., 2021, in which splitter cells could be induced after ~3 plateau potentials. Would simply increasing the learning rate resolve this? If this cannot be replicated, discussion about the disparity should be included.
2. Related, in the same paper it seems as if induced fields did not require further plateau potentials for maintenance. Is this feature also captured by the model?
3. In the same paper all induced cells were initially silent cells, i.e. those that did not have any somatic activity before induction. How would this behavior be explained by the model?

4. Lines 186-188, describing the reward exceeding expectation or not could be made more clear. The expected reward in either task type is 0.5 given a random policy. The result of any given trial is either 0 (incorrect) or 1 (correct). So it is on average that a state pair leads to reward above chance or not.
5. The paragraph in the discussion starting at line 316 is very important, noting that the outside experimenter chose to induce splitters. Splitter cells emerge in these types of task without explicit induction. This section could be expanded to discuss how these representations may be reinforced in a spontaneous manner.
6. The contextual factor β is a function only of the weights onto the apical compartment. This is an intriguing notion that is related to a previous study on intracellular calcium release. But it seems like a strong feature to include in the model and as such would benefit from stronger justification for its inclusion. Currently it is not obvious if this feature is necessary to generate the results.
7. The collapsing of the CA1-EC-CA1 circuit is acceptable for this study, but additional discussion should be included as to if this kind of learning requires recurrent excitatory connectivity, as this would make predictions about the brain regions that could or could not perform this kind of learning. Additionally, it may be worth discussing if the delays generated by a reciprocal loop across different regions might interfere with the timing necessary for plasticity induction.

Minor Points

8. Equation 11 doesn't quite seem to match the description. It seems like the indices are l and m , rather than i indexing over the two possible choices, as is written.
9. Should line 423 read "Similar to that for W_{ij} "?

Reviewer #2 (Remarks to the Author):

The hippocampus is a central hub for many cognitive processes in the brain. Ever since the discovery of spatially selective neurons, the hippocampus has been proposed to support the formation of cognitive maps, which would enable mammals to mentally and physically navigate the world. This theory has been the subject of intense research, from manipulating and recording hippocampal activity with single-cell resolution in rodents (Robinson et al, 2020; Wilson et al, 1994), to imaging in humans (Constantinescu et al, 2016), and to developing a plethora of computational models (Frank et al, 2003; Stachenfeld et al, 2017). However, it is still largely unknown how one can reconcile single neuron physiology (place cells) with the broader cognitive view of the hippocampus as a system making inferences that ultimately supports complex behaviors.

In this manuscript, Cone and Clopath developed a model to better understand how the physical structure of the hippocampal system (connectivity and plasticity rules) may contribute to the generation

of task-dependent cognitive maps. The model includes 3 parts: First, an agent navigates a Y-maze, where a reward is located either at the end of the left, or right corridor, and whose location is signaled upon entering the maze by a set of cues. Second, an action part, instructing the agent to turn left or right at the decision point. And third, a network of 120 neurons representing the hippocampus, with feedforward projections to the action network, recurrent collaterals projections, and feedforward inputs from another 120 neurons representing the external environment. Each hippocampal neuron is itself composed of 3 compartments, a soma, as well as basal and apical “dendrites”.

The authors find that this model was able to produce splitter cells, which are a specific type of activity observed in hippocampal neurons in such memory-guided tasks. Splitter cells first demonstrated that place cells do not solely encode spatial representations but are also part of a context-dependent association (Wood et al, 2000; surprisingly not referenced in the manuscript). The authors also find that splitter neurons emerge as the agent improves performance at the task, and that the synthetic disruption of splitter activity impacts behavior, and vice versa. This is important, because the model learns the structure of the task without a priori knowledge of the state space(s), i.e. the initial association between visual cues predicting the reward’s upcoming location. On one hand, the manuscript is original, as it is the first model to encompass data-driven hippocampal physiology and circuit organization to explain splitter signals. On the other hand, I have concerns about some oversimplified assumptions the authors make, as well as the overall lack of details in the manuscript:

1. First, I am not sure what new predictions the model generates. The authors emphasize “experimentally testable hypotheses” in the abstract but I could not find them in the manuscript, apart from the link between diminished splitter signals and behavioral performance (i.e., error trials) which has already been abundantly established in the field (Ferbinteanu et al, 2003; Pastalkova et al, 2008; Allen et al, 2012; Bahar et al, 2012; Kinsky et al, 2020; just to cite a few).
2. I am curious whether the model would be able to generalize in a task that has an identical structure but with remapped cue/choice associations. For instance, trained with cues A and B leading to choices C and D, and then tested with cues E and F leading to choices C and D, and vice versa. Generalization is a core function of the hippocampus, but I am worried that the architecture of the present model may not encompass that feature, due to the “static” inputs representing the environment that are sent to the “CA1” neurons. According to the current view of the hippocampus, these inputs should be highly plastic, as they are attractor networks representing “working” memory traces of the context, triggered by the initial visual cue, and generating the downstream mixed splitter context/position signals in CA1.
3. It looks as if splitter activity does not really emerge from the model, but is rather stabilized by the design of the network architecture. In that sense, the claim that the model does not need a priori knowledge of the task structure is not entirely true: such knowledge is provided by the induction of **all** the neurons in a specific context. Emergence would rather imply that a fraction of neurons that were never induced would develop a splitter signal. What would happen if only a few cells were induced at

the beginning of the track, or at the end of the track? Is there a preferred location on the track that would trigger the emergence of context-dependent sequences? Also, physiologically, plateau potentials are becoming an increasingly popular mechanism to explain place field formation, but it is still unclear whether they underlie the formation of all the place fields (Priestley et al, 2022).

4. Besides my concern above, the protocol for place field induction is mysterious, as there are not enough details provided in the manuscript. Does induction happen at trial 1? Are all the cells part of context A induced in the same trial? How does their activity look like before induction (and shortly after, see also point #5 below)? I would like to see more explanations and quantification of the mechanisms by which plateau inductions lead to splitter activity.

5. There is little to no quantification of the learning taking place in the model, except for the splitness score shown in Fig. 3. I would be very interested to see 1) how the weight matrix of recurrent collaterals evolves and stabilizes across trials, 2) example of single place field activity, as well as the evolution of population vectors, 3) how these representations are modulated by the dynamics of the weight matrices, and 4) how behavioral errors impact these dynamics.

6. The authors put a lot of efforts into including actual hippocampal physiology, which is fantastic. However, I feel that there are discrepancies with the literature: the co-tuning of apical-soma in CA1 neurons is actually diminished, and not facilitated, by intracellular mechanisms such as ER-mitochondria tethering (O'Hare et al, 2020). But again, I am not suggesting that the authors get rid of their somatic activity equation. Similarly, it is unknown whether plateau potentials update EC-CA1 synaptic weights, here modeled as recurrent collaterals. However, it was demonstrated that they update CA3-CA1 weights, though these synapses are surprisingly not plastic in the current model (also see point #2).

Reviewer #3 (Remarks to the Author):

This paper shows that a biologically realistic learning rule leans cue dependent splitter cell representations, and that the learning of splitter representations precedes behavioural performance, and that biases in either representation of behaviour leads to corresponding biases in behaviour or representation.

I have no issues with the technical details, or the conclusions drawn. All results do however rely on the induction protocol, which external to the network, and so does reduce the generality of this work. The authors are honest about this, but the work would be improved with a more thorough analysis of the induction protocol.

Comments:

1. The induction protocol seems to do all the heavy lifting of this work, with splitter cells seemingly not emerging without it. Where does it come from? Presumably having the right induction protocol is tantamount to having the task solved. So why bother learning the splitter cells too. Is this right? Overall, it would be very helpful to have a more thorough analysis of how relevant the induction protocol is, e.g., what happens with not induction protocol, with random etc... Of course ideally you wouldn't have to artificially add this protocol, and it would all be learned... I'm not asking for this to all happen, but it would really help to provide a better understanding of how important the induction protocol is.

Minor:

1. Use of the word 'states' where it's really neural activity that's being talked about (e.g. lines 167-179 and lots of other places) is confusing for people who are used to the normal definition of a state (i.e. a particular configuration of the world).

2. What about ca3? You find splitter cells there too...

We have formatted this response as follows:

Original remarks from the reviewers in black italics

Our response in black

Quotations from the updated manuscript in blue

Reviewer #1 (Remarks to the Author):

Animals must pair actions to task structure, and the hippocampus has receptive fields related to these actions. But the link between biophysical plasticity mechanisms and the development of task-relevant maps is not clear. The authors construct a computational model with a hippocampal network and an action network and show that the network develops latent structures that are relevant to solving the task, in agreement with experimental data on splitter cells. This work is important because it provides a framework to understand how agents might solve non-Markovian tasks, i.e. tasks in which actions depend on more than just the immediate prior state. This of course has applications in neuroscience but will be of interest to any field involving action selection and agent policy construction. The model is well-described, is grounded by experimental observations, and makes experimentally testable predictions. The alternative models described are modern, and I am not aware of other modeling studies which aim to directly address the question of learning the appropriate state combinations as this manuscript does.

Overall, I am in support of this paper. I would like to see some more comparisons to experimental data as well as justifications for some of the model design choices.

Major Points

1. Figure 3 demonstrates that several hundred trials are needed before splitter cells emerge. This seems at odds with the experimental data presented in Zhao et al., 2021, in which splitter cells could be induced after ~3 plateau potentials. Would simply increasing the learning rate resolve this? If this cannot be replicated, discussion about the disparity should be included.

This is a very important point that we did not address adequately in our previous manuscript. We now have included a new section in the Results, “Slow, population-level integration of task structure enables fast single cell learning” which demonstrates that single cells can be induced after behavioral training in <10 trials. The section is included below:

“The timescales of learning (hundreds or thousands of trials) which we have so far shown far exceed the known timescales of the induction of splitters and place cells in hippocampus. Experimentally, plateau potentials have been shown to generate place fields or splitters after only a handful (<10) of trials^{23–25}. However, in our model we are training the network and the agent’s behavior from scratch. The iterative nature of the cooperation between behavioral and representation learning (i.e., rewarded behaviors improve representations, which thereby increase rewarded behaviors, etc.) is a slow process that dominates the overall timescale of learning in the network. However, this lengthy process of building the population level “cognitive map” then might allow the network to quickly integrate new single units. To test this, we hold out two neurons from the initial phase of learning (one an A-type splitter and one a B-type splitter), keeping their inputs at zero. After training, we stimulate the external inputs to induce the cell, allowing for their weights to undergo plasticity (Neurons 80 and 81 in Figure 5a). During the initial training, as we have previously shown, cells take

hundreds of trials to form splitters, with the majority of cells reaching the “split” criterion during the “zipper” phase occurring from trial ~300-500 (Supplemental Figure 4). However, upon being induced, the holdout neurons initially form generic place fields, before quickly integrating themselves into the latent A- and B-cue sequential structure (neuron 80 shown in Figure 5b). This integration into the “split” representation now only takes ~10 trials (six in the case of neuron 80), in agreement with the timescales of single-cell plasticity observed experimentally (Figure 5c) ²³⁻²⁵. Since the initial evolution of splitters is slow and concurrent with behavior, our model predicts that the few-shot nature of learning splitters (in particular, as opposed to generic place fields) via artificial induction of BTSP is only applicable after, or during the late stages of behavioral learning”.

Figure 1 – Integration of Task Structure Allows for Few-Shot Learning of Context-Specific Cells

a) Left inset, neuron index corresponding to a given location in the track. Right, evolution of population activity on B trials over the course of training, with neurons 80 and 81 held out of learning until trial 750. After the split population has emerged, plateaus are induced in the two held out neurons, and they are quickly integrated into the split population. **b)** Activity of a held-out silent cell (neuron 80), on both A and B trials, relative to the trial of initial induction after behavioral learning. After only six trials, the cell has developed a split representation of the A and B trials, only demonstrating its location specific field after the presentation of cue A. **c)** Rate map of the one of the held-out neurons (neuron 80), centered around its induction trial after behavioral learning (trial 750 in panel a). After induction, the neuron becomes an A-type splitter.

2. Related, in the same paper it seems as if induced fields did not require further plateau potentials for maintenance. Is this feature also captured by the model?

Because of the weight decay included in our learning rule, in the absence of plateau potentials, all the weights will eventually decay to zero and the splitters will disappear, but on a very long timescale. In theory, homeostatic mechanisms could potentially be added on to the model to assist with field maintenance, but we have not included other forms of plasticity in this model.

3. In the same paper all induced cells were initially silent cells, i.e. those that did not have any somatic activity before induction. How would this behavior be explained by the model?

Indeed, this an important behavior to model – our new section “Slow, population-level integration of task structure enables fast single cell learning” and Figure 5 demonstrate how silent cells can be integrated into the network (see our response to point 1).

4. Lines 186-188, describing the reward exceeding expectation or not could be made more clear. The expected reward in either task type is 0.5 given a random policy. The result of any given trial is either 0 (incorrect) or 1 (correct). So it is on average that a state pair leads to reward above chance or not.

Thank you for pointing this out, you are correct – we have changed the language in that section to better reflect that this effect is on average. For example:

“Note that since “splitting” the state representation and/or changing the policy cannot improve average outcomes in the random reward task (average reward will be 0.5), our representations remain as generic place fields in that case”.

5. The paragraph in the discussion starting at line 316 is very important, noting that the outside experimenter chose to induce splitters. Splitter cells emerge in these types of task without explicit induction. This section could be expanded to discuss how these representations may be reinforced in a spontaneous manner.

It is very true that the inclusion of a manual induction protocol leaves open the question of how, in natural learning, the network acquires or samples induction logics. Other works have approximated plateau induction as a random event (not in the same types of models). We have added more explicit references to the induction protocol being manually induced, and also added a few lines regarding the spontaneous reinforcement of representations. (see below):

“The induction protocol is applied for each trial during learning, so for a single trial, half of the population (either the “B-type” or “A-type”, depending on which of the cues is shown) receives a location-specific plateau (Figure 2b)”.

“The combination of these results shows that the “split” induction protocol (Figure 2b) is necessary but not sufficient to form splitters. Instead, the logic of the induction protocol is only integrated into the network if it leads to an underlying representation which improves the behavioral acquisition of reward. Further, since the agent in our simulations starts with a naïve policy (random action selection), the knowledge of whether a given representation will lead to increased rewards is unknown a priori (leading again to our “chicken and egg” problem; one needs splitters to learn the policy, but one needs to learn the policy to know whether to integrate splitters). The combination of these factors makes learning non-trivial, despite the fact that the split induction protocol itself ostensibly “contains” the split logic”.

“Our model considers fixed plasticity protocols, i.e., we as an outside observer choose to induce splitters in this task. While we do show that induction protocols unnecessary for the task are not integrated into the network, our network does not spontaneously generate plateaus or meta-learn the induction protocol. In natural formation of fields in hippocampus, of course, this loop would also be closed, and likely depends on other areas such as the entorhinal cortex³⁷. One can imagine that

through a noisy process, the network might sample potential latent states via one-shot learning to create quick combinations of external inputs/recurrent feedback. If these latent states are useful, perhaps they are reinforced and remain in the state space, remapping otherwise. It remains to be seen how a biophysically plausible model network could learn appropriate plateau induction protocols, task-relevant representations, and policies simultaneously”.

6. The contextual factor β is a function only of the weights onto the apical compartment. This is an intriguing notion that is related to a previous study on intracellular calcium release. But it seems like a strong feature to include in the model and as such would benefit from stronger justification for its inclusion. Currently it is not obvious if this feature is necessary to generate the results.

This is a good point. We have tried to justify that better. We have tried to give some intuition for the beta factor and its necessity in the network, as well as including additional references to better connect it to experimental literature:

“The degree to which the somatic activity is dependent on recurrent, apical feedback is determined by a modulatory factor which depends on the sum of the incoming synaptic weights onto the apical dendrites (see Methods). In practice, this modulatory factor β regulates how much the soma is listening to its dendrites. For small β , the soma is solely a readout of basal activity, while for large β , it reflects the somatic activity is a combination of basal (external) and apical (recurrent) inputs. This function is crucial, since it allows for the soma to express either context-agnostic or context-sensitive representations, depending on the state of the network. While we do not directly model a biophysical process for this compartment-specific modulation, mechanisms of local dendritic inhibition²², branch strength potentiation²³, and intracellular calcium release²⁴ have been shown to modify the influence of dendrites on the soma (or the co-tuning between the two).

7. The collapsing of the CA1-EC-CA1 circuit is acceptable for this study, but additional discussion should be included as to if this kind of learning requires recurrent excitatory connectivity, as this would make predictions about the brain regions that could or could not perform this kind of learning. Additionally, it may be worth discussing if the delays generated by a reciprocal loop across different regions might interfere with the timing necessary for plasticity induction.

As you correctly point out, the assumption that a CA1-EC-CA1 loop could really be synonymous with the direct recurrent collaterals in our model is probably too much of a stretch (not only because of the time delays involved in the reciprocal loop potentially interfering with plasticity, but also the potential of mixing, degradation, or other re-representation of the CA1 signal as it goes through EC). Broadly speaking, we’ve been thinking a lot about the mapping of our model onto the actual circuitry of hippocampus, especially given that BTSP has now been observed in CA3 (Li et al, 2023). Upon more feedback and reflection, we have decided to just call it a generic “hippocampal” population, likely an amalgam of both CA1 and CA3. Ultimately, either choice of label leads to its own imperfect approximation in mapping our model onto the hippocampal circuitry. As for whether this kind of learning requires excitatory recurrent connectivity, we don’t think we can answer that question generally (one might also imagine that context-specific modulation of somatic activity could be occurring due to a disinhibitory circuit that targets specific dendritic compartments). I think all we can really say regarding this issue is that we chose to use excitatory recurrent connectivity in this model, and that choice was sufficient (but maybe not necessary) to model this phenomenon.

Minor Points

8. Equation 11 doesn't quite seem to match the description. It seems like the indices are l and m , rather than i indexing over the two possible choices, as is written.

Thanks, fixed.

9. Should line 423 read "Similar to that for W_{ij} "?

Yes, it's now fixed.

Reviewer #2 (Remarks to the Author):

The hippocampus is a central hub for many cognitive processes in the brain. Ever since the discovery of spatially selective neurons, the hippocampus has been proposed to support the formation of cognitive maps, which would enable mammals to mentally and physically navigate the world. This theory has been the subject of intense research, from manipulating and recording hippocampal activity with single-cell resolution in rodents (Robinson et al, 2020; Wilson et al, 1994), to imaging in humans (Constantinescu et al, 2016), and to developing a plethora of computational models (Frank et al, 2003; Stachenfeld et al, 2017). However, it is still largely unknown how one can reconcile single neuron physiology (place cells) with the broader cognitive view of the hippocampus as a system making inferences that ultimately supports complex behaviors. In this manuscript, Cone and Clopath developed a model to better understand how the physical structure of the hippocampal system (connectivity and plasticity rules) may contribute to the generation of task-dependent cognitive maps. The model includes 3 parts: First, an agent navigates a Y-maze, where a reward is located either at the end of the left, or right corridor, and whose location is signaled upon entering the maze by a set of cues. Second, an action part, instructing the agent to turn left or right at the decision point. And third, a network of 120 neurons representing the hippocampus, with feedforward projections to the action network, recurrent collaterals projections, and feedforward inputs from another 120 neurons representing the external environment. Each hippocampal neuron is itself composed of 3 compartments, a soma, as well as basal and apical "dendrites". The authors find that this model was able to produce splitter cells, which are a specific type of activity observed in hippocampal neurons in such memory-guided tasks. Splitter cells first demonstrated that place cells do not solely encode spatial representations but are also part of a context-dependent association (Wood et al, 2000; surprisingly not referenced in the manuscript). The authors also find that splitter neurons emerge as the agent improves performance at the task, and that the synthetic disruption of splitter activity impacts behavior, and vice versa. This is important, because the model learns the structure of the task without a priori knowledge of the state space(s), i.e. the initial association between visual cues predicting the reward's upcoming location. On one hand, the manuscript is original, as it is the first model to encompass data-driven hippocampal physiology and circuit organization to explain splitter signals. On the other hand, I have concerns about some oversimplified assumptions the authors make, as well as the overall lack of details in the manuscript:

1. First, I am not sure what new predictions the model generates. The authors emphasize "experimentally testable hypotheses" in the abstract but I could not find them in the manuscript, apart from the link between diminished splitter signals and behavioral performance (i.e., error trials)

which has already been abundantly established in the field (Ferbinteanu et al, 2003; Pastalkova et al, 2008; Allen et al, 2012; Bahar et al, 2012; Kinsky et al, 2020; just to cite a few).

Thanks for bringing this to our attention. Our predictions were hidden in the manuscript. To make them explicit, we concatenated a list in the discussion. We have also tried to distinguish between single-trial performance and learned policy regarding our prediction of the codependence of representation and behavior. We also made sure to add the citations above.

Our predictions are the following:

“This evolution of population activity mirrors an evolution in the recurrent weights (Supplemental Figure 1), whereby neurons that represent positions later in the track slowly become more interconnected with same cue-type neurons from earlier in the sequence. Owing to this phenomenon, our model predicts that the split representation emerges first in cells which encode locations nearest to the cue”.

“Our model would therefore predict that the “split” representation (emerging from training on the cue-dependent task) would collapse after a) switching to the random task, or b) switching to random behavior (perhaps through artificially randomizing whether the animal ends in the left or right “location” in a VR environment)”.

“Since the initial evolution of splitters is slow and concurrent with behavior, our model predicts that the few-shot nature of learning splitters (in particular, as opposed to generic place fields) via artificial induction of BTSP is only applicable after, or during the late stages of behavioral learning”.

Regarding the codependence of learning and behavior:

“Experimental studies have also shown that errors in single-trial behavioral performance are correlated with a decrease in task-specific representations in hippocampus during that trial^{35–37}. To examine this effect, we test whether we can induce long-lasting biases in the policy (continued errors of a specific type over many trials) by artificially modifying the underlying representation. To do so, we ablate splitter cells after initial training. Here we consider two types of ablations in our network – general and specific. For the general ablation, we inactivate a fraction of splitters in the network after behavior reaches asymptote, regardless of their corresponding A- or B-type identity. Once the splitters have been removed, the agent’s performance drops in both the A- and B-type trials. However, if we perform an ablation which only targets splitters of a specific identity (in this case a fraction of B-type splitters are inactivated), the agent demonstrates a feature-specific behavioral deficit which lasts across many trials, as it is unable to reach behavioral asymptote specifically on the B-type trials (Figure 4c). Notably, ablation of some splitters can also degrade the surrounding representation, as cells within the population are themselves responsible for the propagation of the cue memory”.

2. I am curious whether the model would be able to generalize in a task that has an identical structure but with remapped cue/choice associations. For instance, trained with cues A and B leading to choices C and D, and then tested with cues E and F leading to choices C and D, and vice versa. Generalization is a core function of the hippocampus, but I am worried that the architecture of the present model may not encompass that feature, due to the “static” inputs representing the environment that are sent to the “CA1” neurons. According to the current view of the hippocampus, these inputs should be highly plastic, as they are attractor networks representing “working” memory

traces of the context, triggered by the initial visual cue, and generating the downstream mixed splitter context/position signals in CA1.

That’s a great question! To test this, we ran new simulations with plastic inputs. We then switched out one of the cues and one of the reward zones, to show that the network can generalize to new inputs which still match the latent structure of the task. Please see the new text and figure below:

“The integration of the task also allows for the network to generalize across environments which map onto the same latent structure. To demonstrate this, we enable input learning (alongside recurrent and behavioral learning) in the cue-dependent task and introduce a novel cue and novel reward port after initial training. We find that the network tracks changes in the input by “relabeling” or “remapping” the existing internal representations onto the new external stimuli. Though the inclusion of input learning increases variance in the representation, it does so without loss of behavioral function, and both the representation and behavior are unaffected by the switch to the novel components of the environment (Supplemental Figure 2)”.

Supplemental Figure 1 – Network Generalizes to Novel External Inputs Which Match the Task Structure

a) Top, schematic of the task before and after novel external inputs (gamma and delta) are introduced. Bottom, since the new inputs do not change the structure of the task, the internal representation of the task should not change after the switch. **b)** Input weights and recurrent weights, before and after the switch. The input weights that synapse onto neurons 11-20 and 101-110 learn to be responsive to a different population of presynaptic input neurons (the population representing the new cues). The

recurrent weights are largely unchanged before and after the introduction of the novel cues. **c)** Evolution of splitness over the course of training. Notably, input learning increases the variance in the neural activity from trial to trial. This is perhaps unsurprising, as now input learning, recurrent learning, and behavioral learning are simultaneously interacting with each other. Nevertheless, the splitness of the representation is maintained after the switch to the novel cues. **d)** Evolution of behavior over the course of training. Behavior is unaffected by the switch to novel cues.

*3. It looks as if splitter activity does not really emerge from the model, but is rather stabilized by the design of the network architecture. In that sense, the claim that the model does not need a priori knowledge of the task structure is not entirely true: such knowledge is provided by the induction of *all* the neurons in a specific context. Emergence would rather imply that a fraction of neurons that were never induced would develop a splitter signal. What would happen if only a few cells were induced at the beginning of the track, or at the end of the track? Is there a preferred location on the track that would trigger the emergence of context-dependent sequences? Also, physiologically, plateau potentials are becoming an increasingly popular mechanism to explain place field formation, but it is still unclear whether they underlie the formation of all the place fields (Priestley et al, 2022).*

This is a very important point, thank you for bringing it to our attention. We address this concern by additional simulations and explanations. (We split the points in the answer below). While we do have the task structure encoded in the learning protocol, our model does not have a priori *states*, i.e. a general and pre-existing conjunctive cells which can support split behavior before learning. By “emergence” or “integration” we explicitly mean the development of these split neural activity states when neural activity is initially “unsplit”. The distinction between protocol and states is important, because having the right induction protocol is *not* tantamount to having the task from Zhao et al 2021 solved (excerpt taken from our edits):

“The combination of these results shows that the “split” induction protocol (Figure 2b) is necessary but not sufficient to form splitters. Instead, the logic of the induction protocol is only integrated into the network if it leads to an underlying representation which improves the behavioral acquisition of reward. Further, since the agent in our simulations starts with a naïve policy (random action selection), the knowledge of whether a given representation will lead to increased rewards is unknown a priori (leading again to our “chicken and egg” problem; one needs splitters to learn the policy, but one needs to learn the policy to know whether to integrate splitters). The combination of these factors makes learning non-trivial, despite the fact that the split induction protocol itself ostensibly “contains” the split logic”. Crucially, in order to match both experimental results (Figure 2e,iii and Figure 2e,vi), our model must assume that memory of the cues is in fact *inaccessible* later on in the track, prior to learning (otherwise, we would develop splitters in both the cue-dependent and random tasks). Instead, in order to form splitters only in the cue-dependent task, the network simultaneously must a) learn to propagate the memory of the cue throughout the track, and b) learn that the “split” state representation of cue memory and current location is beneficial for behavioral outcomes.

Your comment is correct to note that inclusion of a manual induction protocol leaves open the question of how, in natural learning, the network acquires or samples induction logic. Other works have approximated plateau induction as a random event (not in the same types of models).

Our model considers fixed plasticity protocols, i.e., we as an outside observer choose to induce splitters in this task. While we do show that induction protocols unnecessary for the task are not integrated into the network, our network does not spontaneously generate plateaus or meta-learn

the induction protocol. In natural formation of fields in hippocampus CA1, of course, this loop would also be closed, and likely depends on other areas such as the entorhinal cortex³⁷. One can imagine that through a noisy process, the network might sample potential latent states via one-shot learning to create quick combinations of external inputs/recurrent feedback. If these latent states are useful, perhaps they are reinforced and remain in the state space, remapping otherwise. It remains to be seen how a biophysically plausible model network could learn appropriate plateau induction protocols, task-relevant representations, and policies well randomized or self-generated induction of plasticity could be implemented simultaneously.

Regarding a preferred location of induction, we ran new simulations and included new figures which show that the representation “unzips” from the location of the cue to the end of the track (see response to point 5 for the figures in question):

“In the “splitter” task, this means that splitters cannot exist at the end of the track exclusively, as they are reliant on the contiguous memory of the cue. This memory is propagated by the network through the previous splitter cells in the sequence (the existence of which depends on their ability to improve behavior).”

...

“This evolution of population activity mirrors an evolution in the recurrent weights (Supplemental Figure 1), whereby neurons that represent positions later in the track slowly become more interconnected with same cue-type neurons from earlier in the sequence. Owing to this phenomenon, our model predicts that the split representation emerges first in cells which encode locations nearest to the cue”.

Regarding other types of plasticity, you are absolutely correct – considering *only* plateau-based plasticity is a simplification – we have added the following section to the discussion:

Additionally, our model does not consider other forms of plasticity aside from BTSP which likely play an important role during field formation in hippocampus¹⁶. Perhaps the few-shot plasticity of BTSP is combined with more conventional, slower forms of plasticity (Hebbian learning, homeostatic mechanisms, etc.) in the formation and maintenance of these cognitive maps.

4. Besides my concern above, the protocol for place field induction is mysterious, as there are not enough details provided in the manuscript. Does induction happen at trial 1? Are all the cells part of context A induced in the same trial? How does their activity look like before induction (and shortly after, see also point #5 below)? I would like to see more explanations and quantification of the mechanisms by which plateau inductions lead to splitter activity.

To address your concerns, we included more detailed descriptions of the induction protocol in the text:

The induction protocol is applied for each trial during learning, so for a single trial, half of the population (either the “B-type” or “A-type”, depending on which of the cues is shown) receives a location-specific plateau (Figure 2b).

We also ran new simulations and created new figures to demonstrate the dynamic evolution of the splitter population via the induction protocol, including the evolution of single cells (shown below in our response to the next point).

5. There is little to no quantification of the learning taking place in the model, except for the splitness score shown in Fig. 3. I would be very interested to see 1) how the weight matrix of recurrent collaterals evolves and stabilizes across trials, 2) example of single place field activity, as well as the evolution of population vectors, 3) how these representations are modulated by the dynamics of the weight matrices, and 4) how behavioral errors impact these dynamics.

Thank you for the detailed comment, that's a great idea. We've now included a Supplementary Figure 1 showing how the weight matrix evolves over trials (both the full matrix before, during, and after learning, and weights onto a single postsynaptic cell throughout the course of learning). Further, we have added the evolution of population vectors to our main results in Figure 3 and have showed an example of single place field activity during the induction process in a new figure, Figure 5. As for how the representations are modulated by the weight dynamics, we believe Supplementary Figure 1, panels d) and e) demonstrate this. See below for the figures and the captions.

Figure 2 – Behavior and Representation Iteratively Improve Each Other

a) Schematic of representation (left) and agent behavior (right), before (top) and after (bottom) learning the cue-dependent task. Initially, the agent cannot implement the optimal policy at C since a single state cannot incorporate two separate state-action values. After training, the state has been split, and the policy has been learned. **b)** Top, difference in firing rates (activity) on A and B cue trials (“splitness”) of the two populations over the course of learning. Bottom, behavioral performance, shown as the percentage of correct turns over the course of learning. **b)** Agent learns to maximize reward in the task over 4000 trials, with the standard plasticity protocol. Behavioral improvement begins concurrently with the existence of both A and B splitters, both of which are required to solve the task. **c)** Left inset, neuron index corresponding to a given location in the track. Right, evolution of population activity on B trials over the course of training. Odd numbered neurons are induced on B trials, while even numbered neurons are induced on A trials. Splitters emerge in a “zipper”-like fashion, starting from the part of C closest to the cue zone (neuron 30), and propagating along until the end of the track nearest to the reward (neuron 90). This zipping creates two feed-forward “chains” of splitters, one propagating the memory of cue A, and the other propagating the memory of cue B.

Figure 3 – Integration of Task Structure Allows for Few-Shot Learning of Context-Specific Cells

a) Left inset, neuron index corresponding to a given location in the track. Right, evolution of population activity on B trials over the course of training, with neurons 80 and 81 held out of learning until trial 750. After the split population has emerged, plateaus are induced in the two held out neurons, and they are quickly integrated into the split population. **b)** Activity of a held-out silent cell (neuron 80), on both A and B trials, relative to the trial of initial induction after behavioral learning. After only six trials, the cell has developed a split representation of the A and B trials, only demonstrating its location specific field after the presentation of cue A. **c)** Rate map of the one of the held-out neurons (neuron 80), centered around its induction trial after behavioral learning (trial 750 in panel a). After induction, the neuron becomes an A-type splitter.

Supplemental Figure 2 – Recurrent Weights Evolve to Form Splitters

a) Neural index positions in the maze. Index is determined by the input matrix M which assigns external positions to HPC neurons. Neurons 31-90 are candidate splitters representing positions in zone C. **b)** Recurrent weights before training are small and random, meaning somatic activity is not initially context dependent. **c)** During learning, weights begin to “zipper” as the A- and B- related sequences split from each other. For this trial (trial 350 of B), the early neurons in the sequence have already split, but ones close to the end of the track are still acting as generic place fields. **d)** After learning, all the neurons that represent locations within the track have split and receive inputs from other cells that share their cue identity. **e)** Evolution of weights that synapse onto neuron 80. Between trials 300 and 500, neuron 80 begins to selectively receive inputs from previous splitter neurons which

are sensitive to the same cue identity (A). **f**) As the weights evolve, the somatic activity of neuron 80 begins to disappear from B trials, before becoming a complete splitter around trial 500.

6. The authors put a lot of efforts into including actual hippocampal physiology, which is fantastic. However, I feel that there are discrepancies with the literature: the co-tuning of apical-soma in CA1 neurons is actually diminished, and not facilitated, by intracellular mechanisms such as ER-mitochondria tethering (O'Hare et al, 2020). But again, I am not suggesting that the authors get rid of their somatic activity equation. Similarly, it is unknown whether plateau potentials update EC-CA1 synaptic weights, here modeled as recurrent collaterals. However, it was demonstrated that they update CA3-CA1 weights, though these synapses are surprisingly not plastic in the current model (also see point #2).

We appreciate the detailed comment. Regarding our claims about our somatic equation and its relation to intracellular calcium release, if we understood that correctly, we believe it is consistent with the literature, as we did not claim that an *increase* in ER-mitochondria tethering was the basis of the process, but simply stated that manipulations that affect ICR can lead to increased apical co-tuning with the soma (indeed, O'Hare et al. 2022 found that it was through ER-mitochondria *untethering* via which intracellular calcium release and apical co-tuning with the soma may be increased). From our work:

The soma's dynamic sensitivity to apical activity is not based on one particular biophysical process but might be similar to recent experimental findings of the *increased apical co-tuning with the soma following manipulations of intracellular calcium release in vivo*²².

From O'Hare et al, 2022:

The gene *Pdzd8* encodes a recently identified ER-mitochondrial tethering protein that, *when deleted, leads to unrestricted ICR. We found that *Pdzd8* deletion in single adult CA1PNs in vivo substantially increased the level of spatial co-tuning observed in their apical dendrites relative to the soma of CA1 place cells, a phenomenon not observed in basal dendrites, which were already highly co-tuned with the soma in control CA1PNs.*

Broadly speaking, we've been thinking a lot about the mapping of our model onto the actual circuitry of hippocampus, especially given that BTSP has now been observed in CA3 (Li et al, 2023). Upon more feedback and reflection, we have decided to just call it a generic "hippocampal" population, likely an amalgam of both CA1 and CA3. Regarding the reviewer's last point, we have now included a new simulation in which input weights are plastic (see response to point #2).

Reviewer #3 (Remarks to the Author):

This paper shows that a biologically realistic learning rule leans cue dependent splitter cell representations, and that the learning of splitter representations precedes behavioural performance, and that biases in either representation of behaviour leads to corresponding biases in behaviour or representation. I have no issues with the technical details, or the conclusions drawn. All results do however rely on the induction protocol, which external to the network, and so does reduce the generality of this work. The authors are honest about this, but the work would be improved with a more thorough analysis of the induction protocol.

Comments:

1. The induction protocol seems to do all the heavy lifting of this work, with splitter cells seemingly not emerging without it. Where does it come from? Presumably having the right induction protocol is tantamount to having the task solved. So why bother learning the splitter cells too. Is this right? Overall, it would be very helpful to have a more thorough analysis of how relevant the induction protocol is, e.g., what happens with not induction protocol, with random etc... Of course ideally you wouldn't have to artificially add this protocol, and it would all be learned... I'm not asking for this to all happen, but it would really help to provide a better understanding of how important the induction protocol is.

This is a very important point, thank you for bringing it to our attention. In short, having the right induction protocol is **not** tantamount to having the task from Zhao et al 2021 solved (excerpt taken from our edits):

“The combination of these results shows that the “split” induction protocol (Figure 2b) is necessary but not sufficient to form splitters. Instead, the logic of the induction protocol is only integrated into the network if it leads to an underlying representation which improves the behavioral acquisition of reward. Further, since the agent in our simulations starts with a naïve policy (random action selection), the knowledge of whether a given representation will lead to increased rewards is unknown a priori (leading again to our “chicken and egg” problem; we need splitters to learn our policy, but we need to learn our policy to know whether to integrate splitters). The combination of these factors makes learning non-trivial, despite the fact that the split induction protocol itself ostensibly “contains” the split logic. Crucially, in order to match both experimental results (Figure 2e,iii and Figure 2e,vi), our model must assume that memory of the cues is in fact **inaccessible** later on in the track, prior to learning (otherwise, we would develop splitters in both the cue-dependent and random tasks). Instead, in order to form splitters only in the cue-dependent task, the network simultaneously must a) learn to propagate the memory of the cue throughout the track, and b) learn that the “split” state representation of cue memory + current location is beneficial for behavioral outcomes. Note that since “splitting” the state representation and/or policy cannot improve outcomes in the random reward task, so our representations remain as generic place fields in that case”.

Your comment is correct to point out that the inclusion of a manual induction protocol leaves open the question of how, in natural learning, the network acquires or samples induction logic. Other works have approximated plateau induction as a random event (not in the same types of models).

“Our model considers fixed plasticity protocols, i.e., we as an outside observer choose to induce splitters in this task. While we do show that induction protocols unnecessary for the task are not integrated into the network, our network does not spontaneously generate plateaus or meta-learn the induction protocol. In natural formation of fields in hippocampus, of course, this loop would also be closed, and likely depends on other areas such as the entorhinal cortex³⁷. One can imagine that through a noisy process, the network might sample potential latent states via one-shot learning to create quick combinations of external inputs/recurrent feedback. If these latent states are useful, perhaps they are reinforced and remain in the state space, remapping otherwise. It remains to be seen how a biophysically plausible model network could learn appropriate plateau induction protocols, task-relevant representations, and policies simultaneously”.

We also have included a new simulation and supplementary figure to show the results of the network when the induction protocol is completely random. See below:

Supplemental Figure 3 – Random Induction Protocol is Insufficient to Form Splitters

a) Top, difference in firing rates (activity) on A and B cue trials (“splitness”) of the two populations over the course of learning. Bottom, behavioral performance, shown as the percentage of correct turns over the course of learning. Agent fails to maximize reward in the task over 4000 trials, with a random plasticity protocol. **b)** Left inset, neuron index corresponding to a given location in the track. Right, evolution of population activity on B trials over the course of training. Plateau events are induced randomly amongst the neural population (random in both trial identity and location), and the resulting representation fails to split.

“Though random induction alone is insufficient (Supplemental Figure 3), one can imagine that through a noisy process, the network might sample potential latent states via one-shot learning to create quick combinations of external inputs/recurrent feedback”.

Minor:

1. Use of the word ‘states’ where it’s really neural activity that’s being talked about (e.g. lines 167-179 and lots of other places) is confusing for people who are used to the normal definition of a state (i.e. a particular configuration of the world).

We’ve tried to clarify our language here, using “state” in isolation when we mean a particular configuration of the world, and “state representation” when we mean neural activity that is representing a particular configuration of the world. See below for an excerpt from the introduction:

“Agents in these tasks must either pre-assign or learn state representations to create an actionable map of the environment. Animals must similarly represent the state of the environment (or task) via patterns of neural activity (a “state representation”), which can then act as substrates for planning, behavior, memory, etc”.

2. What about ca3? You find splitter cells there too...

Very true! We've been thinking a lot about the mapping of our model onto the actual circuitry of hippocampus, especially regarding recent results. Considering the recurrent connectivity of our model, CA3 is likely a good candidate for our population of hippocampal neurons, given that now BTSP has been observed in CA3 (Li et al 2023). The original results we were attempting to replicate were from CA1, but the representation might be inherited from CA3. Upon more feedback and reflection, we have decided to just call it a generic "hippocampal" population, likely an amalgam of both CA1 and CA3.

REVIEWERS' COMMENTS

Reviewer #1 (Remarks to the Author):

The authors provide a thorough response with new figures which has mostly addressed my concerns. See specifics below

1) Thorough response, thank you.

2) Addressed

3) It is good to see that silent cells can be rapidly integrated into the network, but I was originally after a different question related to the nature of silent cells. Namely, does a plateau potential act to allow spatially selective input that already existed to propagate to the soma (As in Lee, Lin, Lee, 2012) or to induce rapid plasticity? But I can see this is outside the scope of this work, so the response is sufficient.

4) Addressed

5) This is a good expansion, but I think the question of how critical is the specific induction protocol remains. The additional "random induction" figure suggests that completely random plateau potentials will not support split learning. I wonder if there is somewhere in between totally random and completely deterministic the network is able to solve the task.

6) This is a good expansion. Ideally I'd like to see a demonstration of the network's performance at different levels of how sharp the β transition is. The simplest model would have a constant, even weighting of apical and basal inputs, with a value of $m=0$ in equation 4. Can the network learn the task with $m=0$?

7) Addressed

Reviewer #2 (Remarks to the Author):

The authors have done a serious work in addressing all my previous comments. I am glad to see concerns related to the induction protocol and the capability of the model to generalize have been thoroughly addressed. I believe the manuscript is appropriate for publication in Nature Communications.

Reviewer #3 (Remarks to the Author):

I'm happy with the authors' corrections. It's a nice paper.

Reviewer #1 (Remarks to the Author):

The authors provide a thorough response with new figures which has mostly addressed my concerns. See specifics below

1) Thorough response, thank you.

2) Addressed

3) It is good to see that silent cells can be rapidly integrated into the network, but I was originally after a different question related to the nature of silent cells. Namely, does a plateau potential act to allow spatially selective input that already existed to propagate to the soma (As in Lee, Lin, Lee, 2012) or to induce rapid plasticity? But I can see this is outside the scope of this work, so the response is sufficient.

Thank you. In this case, the plateau potential merely induces rapid plasticity.

4) Addressed

5) This is a good expansion, but I think the question of how critical is the specific induction protocol remains. The additional "random induction" figure suggests that completely random plateau potentials will not support split learning. I wonder if there is somewhere in between totally random and completely deterministic the network is able to solve the task.

Indeed, this question remains, and we have added a section to the discussion (see below) to address this.

Though random induction alone is insufficient (Supplemental Figure 4), one can imagine that through a noisy process, the network might sample potential latent states via one-shot learning to create quick combinations of external inputs/recurrent feedback. If these latent states are useful, perhaps they are reinforced and remain in the state space, remapping otherwise. It remains to be seen how a biophysically plausible model network could learn appropriate plateau induction protocols, task-relevant representations, and policies simultaneously. Additionally, our model does not consider other forms of plasticity aside from BTSP which likely play an important role during field formation in hippocampus¹⁶. Likely, the few-shot plasticity of BTSP is combined with more conventional, slower forms of plasticity (Hebbian learning, homeostatic mechanisms, etc.) in the formation and maintenance of these cognitive maps.

6) This is a good expansion. Ideally I'd like to see a demonstration of the network's performance at different levels of how sharp the β transition is. The simplest model would have a constant, even weighting of apical and basal inputs, with a value of $m=0$ in equation 4. Can the network learn the task with $m=0$?

Since beta is a function of the incoming weights, slope of the β transition function essentially creates a dynamic response range stochastic changes in these weights (which arise due to spontaneous behavioral asymmetries). However, as it pertains to the network performance, the impact of the β transition is complicated and also depends on parameters which effect the stochasticity in behavior and the fluctuation size ΔW . In the simplest model case (basal and apical inputs equal), the model

would not form splitters, since the basal component would always be non-zero (i.e. the soma would always represent the unmodified external environment, plus a recurrent feedback term).

7) Addressed

Reviewer #2 (Remarks to the Author):

The authors have done a serious work in addressing all my previous comments. I am glad to see concerns related to the induction protocol and the capability of the model to generalize have been thoroughly addressed. I believe the manuscript is appropriate for publication in Nature Communications.

Reviewer #3 (Remarks to the Author):

I'm happy with the authors' corrections. It's a nice paper.